# Seed Halopriming: A Promising Strategy to Induce Salt Tolerance in Indonesian Pigmented Rice

**DOI:** 10.3390/plants12152879

**Published:** 2023-08-05

**Authors:** Yekti Asih Purwestri, Siti Nurbaiti, Sekar Pelangi Manik Putri, Ignasia Margi Wahyuni, Siti Roswiyah Yulyani, Alfino Sebastian, Tri Rini Nuringtyas, Nobutoshi Yamaguchi

**Affiliations:** 1Department of Tropical Biology, Faculty of Biology, Universitas Gadjah Mada, Yogyakarta 55281, Indonesia; siti.nurbaiti@mail.ugm.ac.id (S.N.); tririni@ugm.ac.id (T.R.N.); 2Research Center for Biotechnology, Universitas Gadjah Mada, Yogyakarta 55281, Indonesia; 3Biotechnology Master Program, The Graduate School, Universitas Gadjah Mada, Yogyakarta 55281, Indonesia; sekarpelangi.mp@mail.ugm.ac.id (S.P.M.P.); ignasiamargi02@mail.ugm.ac.id (I.M.W.); sitiryulyanims@mail.ugm.ac.id (S.R.Y.); 4Institute of Plant Science and Resources, Okayama University, Okayama 710-0046, Japan; alfino-sebastian@s.okayama-u.ac.jp; 5Plant Stem Cell Regulation and Floral Patterning Laboratory, Graduate School of Science and Technology, Nara Institute of Science and Technology, Ikoma 630-0101, Japan; nobuy@bs.naist.jp

**Keywords:** seed halopriming, salt tolerance, physiological responses, gene expression, pigmented rice

## Abstract

Unfavorable environmental conditions and climate change impose stress on plants, causing yield losses worldwide. The Indonesian pigmented rice (*Oryza sativa* L.) cultivars Cempo Ireng Pendek (black rice) and Merah Kalimantan Selatan (red rice) are becoming popular functional foods due to their high anthocyanin contents and have great potential for widespread cultivation. However, their ability to grow on marginal, high-salinity lands is limited. In this study, we investigated whether seed halopriming enhances salt tolerance in the two pigmented rice cultivars. The non-pigmented cultivars IR64, a salt-stress-sensitive cultivar, and INPARI 35, a salt tolerant, were used as control. We pre-treated seeds with a halopriming solution before germination and then exposed the plants to a salt stress of 150 mM NaCl at 21 days after germination using a hydroponic system in a greenhouse. Halopriming was able to mitigate the negative effects of salinity on plant growth, including suppressing reactive oxygen species accumulation, increasing the membrane stability index (up to two-fold), and maintaining photosynthetic pigment contents. Halopriming had different effects on the accumulation of proline, in different rice varieties: the proline content increased in IR64 and Cempo Ireng Pendek but decreased in INPARI 35 and Merah Kalimantan Selatan. Halopriming also had disparate effects in the expression of stress-related genes: *OsMYB91* expression was positively correlated with salt treatment, whereas *OsWRKY42* and *OsWRKY70* expression was negatively correlated with this treatment. These findings highlighted the potential benefits of halopriming in salt-affected agro-ecosystems.

## 1. Introduction

Rice (*Oryza sativa* L.) is a major source of carbohydrates and a staple food across Asia [1]. Various rice cultivars have pigmented grains containing anthocyanins and other phenolic compounds, which provide health benefits due to their antioxidant, inflammatory, and anticancer properties [2,3,4,5]. Pigmented rice is gaining in popularity and is widely consumed as a functional food. Many local pigmented rice cultivars with red and black grains are grown in Indonesia [6,7]. Unfortunately, like most crop plants, rice is a glycophyte, i.e., it is sensitive to salinity, particularly during the early seedling and reproductive stages [8]. Indeed, rice is even more sensitive to high-salinity conditions than other cereals such as barley (*Hordeum vulgare* L.), wheat (*Triticum aestivum* L.), and rye (*Secale cereale* L.) [9]. However, although soil salinization is an increasing problem worldwide, few studies have focused on the potential cultivation of pigmented rice in marginal, high-salinity land.

Salt stress can reduce rice production to almost zero and reduce dry biomass accumulation by up to 90% in susceptible cultivars, especially during the dry season [10]. Salt stress limits water availability and leads to osmotic stress, sodium accumulation resulting in ion toxicity, and nutritional and phytohormonal imbalance, ultimately affecting plant growth and development [11]. Ionic and oxidative stresses due to salt stress cause cell damage, senescence, and even leaf death, reducing the supply of energy to the plant [12].

Plants acclimate to unfavorable environments by altering their morphology, physiology, and biochemistry; these responses are driven by the activation of specific genes [13]. Plants also use enzymatic and non-enzymatic antioxidant machineries to protect their cells from the harmful effects of cytotoxic free radicals, which include the enzymes superoxide dismutase (SOD), catalase (CAT), and guaiacol peroxidase (GPX), as well as reduced glutathione (GSH) [14]. Several genes encoding antioxidant enzymes play important roles in rice under salt stress, including *CuZnSOD*, cytosolic ascorbate peroxidase (*cytAPX*,) and *CAT* [15,16]. For instance, the expression of *CuZnSOD1*, *APX2*, and *CAT3* increases in rice leaves experiencing salt stress [17]. Apart from antioxidant activity, plants respond to stress conditions through osmotic adjustment by producing and accumulating compatible solutes such as proline to maintain the structure and integrity of the cell wall [18,19]. Proline is a highly water-soluble amino acid that helps plants adjust the osmotic potential of cells. *P5CS2* and *P5CR* are involved in proline biosynthesis, encoding enzymes in the glutamate pathway (∆1-pyrroline-5-carboxylate synthase and pyrroline-5-carboxylate reductase, respectively). *P5CS1* encodes a protein that controls the biosynthesis of proline, whose expression increases under stress conditions, in the chloroplast [20].

Numerous transcription factors (TFs) regulate the expression of genes encoding functional proteins [13]. Any minor change in TF regulation, their sequences, or their target DNA sequences greatly alters gene regulatory networks and, thus, plant physiology or morphology. Many TFs are involved in regulating plant defense, with MYB and WRKY TF family members playing important roles in plant responses to salt stress [21,22,23,24,25,26,27,28]. *OsMYB91* is induced by salt stress, and rice plants overexpressing *OsMYB91* showed increased salt tolerance, enhanced capacity to scavenge reactive oxygen species (ROS), increased induction of *OsP5CS1* expression, and greater proline accumulation compared to the wild type [27]. Various *WRKY* genes are expressed in rice plants under salinity-stress conditions, including *OsWRKY42* and *OsWRKY70* [25,27]. OsWRKY42 and OsWRKY70 are thought to negatively regulate salinity tolerance because they are involved in leaf senescence [29] and alleviate osmotic stress tolerance [30].

Improving salt tolerance in rice is a promising strategy to overcome yield reductions caused by salt stress [31]. Seed priming, or pre-germinative seed treatment (including hydropriming, halopriming, osmopriming, hardening, and phytohormonal priming), is used to improve crop performance by enhancing seed germination rates, seed uniformity, and stress tolerance [20,32]. Seed priming has several advantages, as it increases germination at a uniform rate, is easy to apply, and is inexpensive and low risk [33,34]. Priming provides seeds with stress memory, which can be recruited upon subsequent stress exposure by triggering the activation of genes, antioxidant compounds, and TFs that increase plant tolerance to salt stress [32,35,36]. The halopriming of seeds helps repair damaged DNA to prevent damage to the template used for replication and transcription. Halopriming with CaCl_2_ or KCl improves salt tolerance by reducing osmotic stress and Na+ uptake and increasing K+ uptake to maintain chlorophyll accumulation and nutrient partitioning [36,37,38].

We previously demonstrated that seed halopriming improved salt tolerance in Indonesian pigmented rice by inducing the expression of genes encoding the transporters responsible for maintaining ionic balance [39]. In the current study, we further investigated the effects of seed halopriming in response to salt stress in two Indonesian pigmented rice cultivars (Cempo Ireng Pendek and Merah Kalimantan Selatan, a black and a red rice, respectively), the salt-sensitive cultivar IR64, and the salt-tolerant cultivar INPARI 35, under normal and salt-stress conditions, focusing on proline, TF encoded stress-related genes (*OsMYB91, OsWRKY42, OsWRKY70*), and antioxidant-enzyme-related genes. Our findings shed light on the effects of halopriming in rice, particularly nutrient-rich, Indonesian pigmented rice.

## 2. Results

### 2.1. Effects of Seed Halopriming on Germination

The germination rates of all primed and non-primed cultivars were similar, reaching 100% on the fourth day of soaking, except for Merah Kalimantan Selatan, which reached 100% germination after only 2–3 days of soaking (Table 1). The mean germination time (MGT) markedly increased in response to seed halopriming in all rice cultivars except INPARI 35 compared to that of non-primed control seeds, as shown in Table 1. Cempo Ireng Pendek had the longest germination time, with its the earliest germination observed ~2 days after the onset of soaking. We observed a delayed germination time under the halopriming treatment in all cultivars except the salt-tolerant INPARI 35, but halopriming did not decrease the number of seeds that ultimately germinated.

### 2.2. Plant Growth and Relative Water Content (RWC)

To explore the effects of halopriming on seedling growth in rice under salt stress, we measured the plant height and the root length as indicators of plant growth. The seedling height and especially the root length generally decreased in response to salt-stress treatment (Table 2) with significantly reduced root length seen in all cultivars except Merah Kalimantan Selatan. However, the plant height and root length did not significantly differ between the primed and non-primed seedlings in any cultivar.

Salt stress significantly reduced the RWC in both non-primed and haloprimed seedlings. Under salt stress, halopriming treatment had no significant effect on the RWC compared to no priming in all cultivars, except for IR64. Halopriming in IR64 resulted in a significantly higher RWC than in non-primed plants (8.1%). The strongest reduction in RWC under salt-stress conditions was observed in Merah Kalimantan Selatan, which reached 13.25% and 16.15% under non-primed and haloprimed conditions, respectively, compared to the control.

### 2.3. Total ROS Contents and Membrane Stability Index

We examined ROS accumulation and the membrane stability index as measures of the oxidative damage caused by salt stress in rice. Different rice cultivars exhibited different levels of ROS accumulation after 7 days of salt treatment (Figure 1a). The salt-sensitive IR64 showed two times higher ROS contents under salt stress compared to non-stress conditions in both non-primed and haloprimed seedlings, up to 91.76 L/mol and 76.33 L/mol, respectively. Halopriming of IR64 and Cempo Ireng Pendek seeds resulted in a significantly lower ROS accumulation under salt stress compared to that in non-primed seedlings (15.33 L/mol cm and 13.78 L/mol cm). We observed no significant differences in ROS levels for Merah Kalimantan Selatan under salt-stressed and non-stressed conditions in either haloprimed or non-primed seedlings.

Moreover, salt stress sharply decreased the membrane stability index in three of the four cultivars examined (Figure 1b). Under saline conditions, halopriming led to a much higher membrane stability in IR64, Cempo Ireng Pendek, and Merah Kalimantan Selatan, though it had no detectable effect in the salt-tolerant INPARI 35. The improvement of the membrane stability index reached approximately two-fold in IR64 and Cempo Ireng Pendek. Higher ROS levels under salt stress in IR64 corresponded to a strong decrease in the membrane stability index, from 91.45% to 9.57% in non-primed seedlings and 87.92% to 22.22% in haloprimed seedlings under normal compared to high-salinity conditions, respectively. Haloprimed Cempo Ireng Pendek had the highest membrane stability and the lowest ROS accumulation under salt stress.

### 2.4. Photosynthetic Pigment Contents

Seedlings exposed to salt stress, with or without prior halopriming, had significantly lower total chlorophyll contents than seedlings grown under non-stress conditions (Figure 2a). The total chlorophyll contents of non-primed IR64 and Merah Kalimantan Selatan seedlings decreased by almost 52% under salt-stress conditions compared to decreases of 39% and 23% in haloprimed plants, respectively. Under salt stress, the total chlorophyll contents of haloprimed IR64 and Merah Kalimantan Selatan were significantly higher than those of non-primed seedlings (by 2.37 mg/L and 5.38 mg/L, respectively). By contrast, we did not detect a significant effect of halopriming on the total chlorophyll content in INPARI 35 or Cempo Ireng Pendek.

The different rice cultivars showed different effects on the chlorophyll a/b ratio in response to salt stress (Figure 2b). INPARI 35 and Cempo Ireng Pendek showed slight but not significant decreases in this ratio in non-primed seedlings under stress relative to non-stress conditions (0.18 and 0.25, respectively) but significant decreases in haloprimed seedlings (1.31 and 0.55, respectively). IR64, a susceptible cultivar, showed a significant increase in the chlorophyll a/b ratio in non-primed seedlings, but not in haloprimed seedlings, under salt stress compared to non-stress conditions (0.87). Merah Kalimantan Selatan showed no significant differences in the chlorophyll a/b ratio under stress and non-stress conditions in either non-primed or haloprimed seedlings.

In addition, salinity lowered the carotenoid contents in all cultivars (Figure 2c). All cultivars except Merah Kalimantan Selatan also showed lower carotenoid contents after halopriming compared to no priming under salt stress. The carotenoid contents of IR64 and INPARI 35 decreased by 0.87 mg/L and 1.33 mg/L under salt-stress treatment in haloprimed compared to non-primed seedlings, respectively. Haloprimed Merah Kalimantan Selatan seedlings had similar carotenoid contents regardless of salt stress. Their carotenoid content was 1.47 mg/L higher in haloprimed than in non-primed seedlings under salt stress and decreased by 1.99 mg/L in non-primed seedlings but only by 0.19 mg/L in haloprimed seedlings experiencing salt-stress conditions.

### 2.5. Proline Contents

In this study, all seedlings exposed to salt stress showed significantly higher proline contents than the non-salt-stressed controls for both haloprimed and non-primed seedlings, except for INPARI 35 in which only the non-primed seedlings accumulated more proline (Figure 3). Cempo Ireng Pendek and Merah Kalimantan Selatan subjected to salt stress accumulated up to 5- to 8-fold more free proline than non-stressed seedlings. We observed different changes in proline contents under salt stress in haloprimed seedlings. Indeed, the proline content of IR64 and Cempo Ireng Pendek was greater in haloprimed than in non-primed seedlings (up to 0.16 µg/g FW and 0.24 µg/g FW, respectively). By contrast, in INPARI 35 and Merah Kalimantan Selatan, non-primed, stressed seedlings had higher proline contents than haloprimed seedlings (0.13 µg/g FW and 0.28 µg/g FW, respectively).

### 2.6. Stress-Induced Gene Expression

#### 2.6.1. Expression of Proline Biosynthesis Genes

The *OsP5CS1* expression increased during salt treatment, peaking at 24 h into treatment. We noticed this effect in both haloprimed and non-primed seedlings for all cultivars (Figure 4). After 24 h of salt treatment, the haloprimed seedlings of all cultivars except IR64 had lower *OsP5CS1* expression levels when compared to non-primed seedlings. In INPARI 35 (a salt-tolerant cultivar), *OsP5CS1* was expressed at significantly higher levels in non-primed (6.2-fold) than in haloprimed seedlings (4.3-fold). However, in IR64 (a salt-susceptible cultivar), the *OsP5CS1* transcript levels after 24 h of salt treatment were significantly higher in haloprimed than in non-primed seedlings, with increases of 3.0- and 2.5-fold, respectively. After 24 h of salt treatment, Cempo Ireng Pendek and Merah Kalimantan Selatan appeared to have lower *OsP5CS1* mRNA levels in haloprimed relative to non-primed seedlings, but these differences were not significant.

#### 2.6.2. Expression of Antioxidant Genes

The expression levels of antioxidant genes are shown in Figure 5. In general, antioxidant genes were upregulated after 24 h of salt stress. The seed halopriming treatment did not markedly affect the expression of antioxidant genes compared to that in non-primed seedlings, except for *CuZnSOD1* expression in Cempo Ireng Pendek at 24 h and *APX2* expression in IR64 at 0 h of treatment. *CAT3* had the lowest expression level among the antioxidant genes we assessed. *APX2* was expressed at the highest level in IR64, increasing up to 14-fold after 24 h of salt stress. *CuZnSOD1* was expressed at higher levels (13-fold) than *APX2* (6-fold) in INPARI 35 in response to salt stress. In Cempo Ireng Pendek, both *CuZnSOD1* and *APX2* were upregulated up to 12-fold, in non-primed seedlings upon salt exposure. Unlike other cultivars, Merah Kalimantan Selatan showed relatively low *CuZnSOD1* and *APX2* expression levels, with no significant differences between control and salt stress, but *CAT3* was upregulated by 3.3- and 5.5-fold in non-primed and haloprimed seedlings, respectively.

#### 2.6.3. Expression of TF-Related Genes

In this study, the *OsMYB91* expression increased after 24 h of salt treatment with or without halopriming in all cultivars except IR64 (Figure 6). The salt-tolerant cultivar INPARI 35 showed the highest *OsMYB91* expression level, which increased 5.2- and 13.8-fold after 24 h of salt treatment for non-primed and haloprimed seedlings, respectively. Similarly, the *OsMYB91* expression increased 2.7-fold in haloprimed relative to non-haloprimed Cempo Ireng Pendek seedlings after 24 h of salt treatment and 1.4-fold in Merah Kalimantan Selatan. By contrast, in the salt-stress-sensitive cultivar IR64, *OsMYB91* showed the lowest expression level but relatively high increases of 4.8- and 2.7-fold under salt stress in non-primed and haloprimed seedlings, respectively.

Figure 7 shows the expression patterns of the *WRKY* genes. In IR64 and INPARI 35, *OsWRKY42* and *OsWRKY70* were generally expressed at lower levels in haloprimed compared to non-primed seedlings, but they were expressed at higher levels in some Cempo Ireng Pendek and Merah Kalimantan Selatan seedlings after halopriming. Under salt stress, the *OsWRKY42* expression gradually decreased in all cultivars, except for non-primed Merah Kalimantan Selatan seedlings, which showed the highest *OsWRKY42* expression level after 6 h of salt stress (a two-fold increase). The salt-tolerant cultivar INPARI 35 had the lowest *OsWRKY42* expression. *OsWRKY70* showed different expression patterns in different cultivars under salt treatment. Seed halopriming significantly reduced the expression of *OsWRKY70* compared to that in non-primed seedlings in Cempo Ireng Pendek. We observed similar expression patterns in IR64 (at 0 and 24 h) and INPARI 35 (at 24 h). The highest *OsWRKY70* expression level was in non-primed Cempo Ireng Pendek seedlings, in which it increased 3.5-fold after 6 h of salt stress. Seed halopriming had the strongest effect on the *OsWRKY70* expression in Cempo Ireng Pendek but did not affect the expression of this gene in Merah Kalimantan Selatan.

## 3. Discussion

Seed priming as a pre-germinative treatment might create a moderate stress signal that elicits a “priming memory” to help plants cope with subsequent stress and improve stress tolerance [40]. In this study, we determined that halopriming treatment extended the duration of seed germination in all four rice cultivars examined. Treatment with a halopriming solution decreases water uptake by seeds, thus delaying the initiation of the growth phase and extending the activation phase of the seed, during which chromosome and DNA damage repair, early DNA replication, and protein synthesis take place [35,41]. The specifics of the priming treatment, including the treatment composition and concentration, and the priming duration are important factors that help determine germination success and seedling establishment. Based on the current results, we conclude that our halopriming technique did not have harmful effects on the rice seeds, as all four cultivars reached 100% seed germination.

All four cultivars were severely injured by salt stress, as indicated by the higher ROS level and the decreasing total chlorophyll content, carotenoid content, membrane stability, and RWC of the seedlings. Excessive ROS production leads to oxidative stress and damages the cell membrane, proteins, and chlorophyll, ultimately leading to cell death [42,43]. Salt stress also led to increases in the accumulation of proline and in the expression levels of the proline biosynthesis gene *OsP5CS1,* genes encoding antioxidant enzymes, and the TF gene *OsMYB91* in all cultivars.

Halopriming increased the salt tolerance in Indonesian pigmented rice cultivars, evoking different responses that contribute to the varying levels in the salt tolerance of the cultivars. Halopriming significantly affected almost all parameters in the salt-sensitive cultivar IR64 but had no significant effect on the salt-tolerant cultivar INPARI 35, except regarding *OsMYB91* and Os*WRKY70* expression. Halopriming strongly affected the black rice Cempo Ireng Pendek, as shown by it showing the lowest ROS levels among the four cultivars, the highest membrane stability index and proline accumulation, and significantly increased *OsMYB91* and *OsWRKY70* expression. Halopriming in the red rice Merah Kalimantan Selatan had less extensive effects but helped maintain photosynthetic pigments and suppressed Os*WRKY42* expression to enhance salt tolerance.

Halopriming led to greater membrane stability. Lower ROS accumulation in the haloprimed plants indicates that priming is beneficial in counteracting stress-induced ROS formation. As a result, the membrane stability index was higher in haloprimed than in non-primed seedlings. Previous studies have demonstrated that seed priming inhibits ROS accumulation, improves membrane stability, and enhances defense mechanisms and stress tolerance [43,44,45]. Based on our results (Figure 1a), we suggest that, in plants that are tolerant of saline conditions, ROS homeostasis is maintained, which helps decrease oxidative damage during salt stress.

Haloprimed seedlings showed higher total chlorophyll contents than non-primed seedlings, especially in IR64 and Merah Kalimantan Selatan. We only observed significantly higher carotenoid contents in haloprimed seedlings in Merah Kalimantan Selatan. The difference in the chlorophyll a/b ratio following halopriming was highly significant in IR64 and INPARI 35. Under salt stress, the efficiency of the photosynthetic system is generally maintained by converting chlorophyll b into chlorophyll a, leading to an increase in the chlorophyll a/b ratio [46]. Importantly, we observed such an increase under salt stress only in IR64, and halopriming sharply decreased this ratio. Higher photosynthetic pigment contents in haloprimed plants have been reported in many studies [35,40,41].

Plants accumulate soluble solutes as a means of enhancing stress tolerance. Proline is an important osmolyte for osmotic adjustment in plants under stress conditions [47]. Proline accumulation represents an acclimation response to stress, as proline helps to balance osmotic pressure in the cell (cell turgor), bring ROS concentrations within normal ranges (preventing oxidative bursts), stabilize cellular membranes (minimizing electrolyte leakage), conserve energy, and protect macromolecules, resulting in improved stress tolerance [42,43,44]. Notably, Merah Kalimantan Selatan showed lower proline contents in haloprimed than in non-primed seedlings. The increase in proline contents we observed might be attributable to strategies adapted by the susceptible cultivar IR64 to cope with salt stress after halopriming treatment. By contrast, it appears that haloprimed Merah Kalimantan Selatan seedlings do not employ proline accumulation as a defense mechanism under stress conditions. Proline accumulates to higher levels after salt stress in parallel with higher *OsP5CS1* expression.

The ability of plants to resist abiotic stress is inseparable from the expression of TF-regulated stress-related genes [48]. The activation or repression of TFs is associated with stress memory induced by initial mild stress priming [49]. Analyzing gene expression patterns provides important clues about the underlying mechanisms.

The role of OsMYB91 as a positive regulator of salt-stress tolerance in plants was reported by Zhu et al. [27]. Furthermore, we observed that halopriming significantly upregulated *OsMYB91* expression (Figure 6). Halopriming also resulted in a significantly lower expression of Os*WRKY70* compared to that in non-primed seedlings after salt stress in all cultivars except Merah Kalimantan Selatan, whereas we observed no significant difference in Os*WRKY42* expression. Han et al. [50] reported that overexpression of *OsWRKY42* led to higher oxidation stress by repressing ROS-scavenging-related gene expression and caused a decrease in the chlorophyll content. Li et al. [30] reported that WRKY70 negatively regulates stomatal conductance, as transgenic plants overexpressing *WRKY70* showed increased susceptibility to osmotic stress, with a higher stomatal conductance that resulted in greater water loss and electrolyte leakage. Here, the low expression levels of *OsWRKY42* and *OsWRKY70* were accompanied by a higher chlorophyll content and a higher membrane stability index. Therefore, we propose that halopriming positively regulates the rice TF gene *OsMYB91* but negatively regulates *OsWRKY42* and *OsWRKY70* in response to salt stress.

Halopriming activates defense mechanisms against salt stress, such as antioxidant defense systems and osmotic adjustment [51]. Seed priming increases the salinity tolerance by accumulating and activating dormant signaling molecules and TFs to control gene expression and enzyme activity [43]. We observed a primed state induced by halopriming pre-treatment in IR64, Cempo Ireng Pendek, and Merah Kalimantan Selatan, which generated stress-like conditions even when the seedlings were growing under normal conditions. These responses included a higher chlorophyll a/b ratio, higher RWC, and lower *OsWRKY70* expression in IR64; higher ROS accumulation and lower *OsWRKY70* expression in Cempo Ireng Pendek; and lower total chlorophyll content in Merah Kalimantan Selatan. These early responses indicate that the seedlings acquired a “stress memory” that primed their response to a subsequent salt-stress condition, allowing them to respond more rapidly and strongly while enhancing their salt tolerance.

In conclusion, given its benefits demonstrated here, seed halopriming could serve as a suitable strategy to increase salt tolerance and enhance plant growth in unfavorable environments due to climate change, which poses great challenges for crop production. One case in which this strategy had been employed was reported by Sembiring [52], who planted salt-tolerant cultivars in coastal areas of Indonesia (on 7–10 ds/m soil salinity level, equal to 70–100 mM NaCl treatment) and obtained a higher rice yield. However, further studies under natural field conditions are needed to gain more insights into the effects of seed halopriming on yield components in rice.

## 4. Materials and Methods

### 4.1. Seed Halopriming

Seeds of the rice cultivars Cempo Ireng Pendek, Merah Kalimantan Selatan, IR64, and INPARI 35 from the Research Center of Biotechnology and the Indonesia Center for Rice Research were used in this study. INPARI 35 is classified as salt-stress tolerant, and IR64 is salt-stress sensitive. The experiments were performed at the Genetic Engineering Laboratory, Research Center of Biotechnology, Universitas Gadjah Mada, Yogyakarta, Indonesia. The seeds were surface sterilized in 10% (*w*/*v*) sodium hypochlorite (NaClO) for 15 min and rinsed three times with distilled water for 15 min each time. For halopriming, the surface-sterilized seeds of each cultivar were soaked for 48 h in a priming solution containing 100 mM NaCl, 2.2% (*w*/*v*) CaCl_2_, 2.2% (*w*/*v*) KCl, 2.2% (*w*/*v*) KNO_3_, and 50 mM H_2_O_2_ [53]. For the control (no priming treatment), seeds were soaked in distilled water. After priming, the seeds were dried at room temperature to their original weight.

### 4.2. Seed Germination

Haloprimed and non-primed seeds were germinated on filter paper moistened with distilled water in Petri dishes. For the germination test, twenty seeds were placed into a Petri dish (9 cm in diameter) with double layers of filter paper, and 10 mL distilled water was pipetted into each Petri dish. Five replicates of the Petri dish were used. Seed germination was recorded daily for seven days, and a seed was considered to have germinated when the emerged radicle measured about 2 mm [54]. The mean germination time (MGT) and the germination rate (GR) were measured after 7 days [55].
MGT (days) = Σ (number of seeds germinated each day × number of days from the beginning of the test)/total number of germinated seeds
GR (%) = (number of seeds germinated/total number of seeds) × 100

### 4.3. Salt-Stress Treatment and Phenotypic Assessment

After 7 days of germination, the seedlings were transferred to a greenhouse in a floating hydroponic cultivation system with trays containing 4 L of Yoshida’s nutrient solution. Each container contained four cultivars of haloprimed and non-primed seedlings. After 2 weeks, the seedlings were subjected to salt stress created by the addition of 150 mM NaCl to the cultivation solution followed by incubation for 7 days. The growth parameters plant height and root length were measured after 7 days of salt-stress treatment. The relative water content (RWC) was measured on day 7 of the salt treatment using the method of Smart and Bingham [56]. The leaves were weighed immediately after harvesting to measure the fresh weight (FW) and soaked in distilled water for 4 h to measure the turgid weight. The samples were dried in an oven at 80 °C for 24 h, and the dry weight was obtained. The RWC was calculated as follows:RWC (%) = [(fresh weight − dry weight)/(turgid weight − dry weight)] × 100

### 4.4. Quantification of Chlorophyll and Carotenoid Contents

The contents of chlorophyll a, chlorophyll b, total chlorophyll, and carotenoids were determined on day 7 of the salt treatment, as described by Harborne [57]. Briefly, 100 mg FW of leaf tissue was ground in liquid nitrogen, extracted in 10 mL of 80% (*v*/*v*) chilled aqueous acetone, and centrifuged at 7871× *g* for 5 min at 4 °C. The absorbance was recorded at different wavelengths (470 nm, 663 nm, and 645 nm) using a UV–visible spectrophotometer VWR UV-1600PC Spectrophotometer (VWR International Europe, Leuven, Belgium). The pigment contents were calculated as follows:Chlorophyll a (mg/L) = (12.21 A_663_ − 2.81 A_645_) × (*w*/*v*)
Chlorophyll b (mg/L) = (20.13 A_645_ − 5.03 A_663_) × (*w*/*v*)
Total chlorophyll (mg/L) = chlorophyll a + chlorophyll b
Carotenoids (mg/L) = [(1000 × A_470_) − (3.27 × chlorophyll a) − (104 × chlorophyll b)]/227

A = absorbance, W = fresh weight of sample (mg), and V = volume (L).

### 4.5. Analysis of Total ROS Contents

Analysis of the ROS contents was carried out as described by Alexou [58] with modifications. Leaf samples were collected on day 7 of the salt treatment. The samples were immediately frozen in liquid nitrogen and pulverized. Each 0.05 g pulverized sample was placed into a microtube and combined with polyvinylpyrrolidone (1:1 *w*/*w*) in 500 μL of distilled water. The mixture was centrifuged at 15,000× *g* at 4 °C for 20 min. The supernatant was collected and centrifuged again at 15,000× *g* at 4 °C for 5 min. A 100 μL aliquot of this second supernatant was transferred to a new microtube and combined with 580 μL of distilled water and 20 μL of 3 mM luminol. The absorbance of this mixture was measured using a UV–visible spectrophotometer at 425 nm and compared to that of a blank mixture of 680 μL of distilled water and 20 μL of 3 mM luminol.

The absorbance results were used to determine the absorbance coefficient (e) with the following formula:e = absorbance/[luminol]

### 4.6. Analysis of Membrane Stability Index

A 200 mg leaf sample was cut into pieces, placed in a tube with 20 mL of ddH_2_O, and incubated at room temperature. After 12 h, the conductivity value of the solution was determined using an EC meter (Electro-conductivity meter), as EC1. The sample was incubated in a 100 °C water bath for 15 min and allowed to cool. After the solution reached a temperature of 25 °C, the conductivity of the solution was measured as EC2 [59].
Membrane stability index (%) = [1 − (EC1/EC2)] × 100

### 4.7. Quantification of Proline Content

The proline contents were measured on day 4 of the salt treatment using the Bates et al. [60] method with modifications [39]. The leaf tissue (0.25 g) was homogenized in 5 mL of 30% (*v*/*v*) sulfosalicylic acid, and the homogenate was filtered through a Whatman No. 2 filter paper. A 1 mL sample of the extract was mixed with 1 mL of glacial acetic acid and 1 mL of acid ninhydrin (1.25 g ninhydrin warmed in 30 mL glacial acetic acid and 20 mL 6 M phosphoric acid until dissolved) for 1 h at 95 °C. The reaction mixture was extracted with 2 mL of toluene. The chromophore-containing toluene was collected, and the absorbance was read at 520 nm. The amount of proline was determined from a standard curve and presented as µmol g^−1^FW.

### 4.8. Total RNA Isolation and RT-qPCR for Gene Expression Analysis

The total RNA was extracted from the leaves of the control and salt-stressed seedlings using a FavorPrep^TM^ Plant Total RNA Mini Kit (Favorgen Biotech Corp., Ping-Tung, Taiwan) following the manufacturer’s instructions. The amount of RNA was measured using a MN-913A MaestroNano^®^ Pro spectrophotometer (Maestrogen Inc., Hsinchu City, Taiwan). First-strand cDNA was synthesized using a ReverTra Ace qPCR RT Master Mix with gDNA Remover (Toyobo, Osaka, Japan). A CFX96^TM^ Real-Time PCR System (Bio-Rad, Singapore) and an ExcelTaq^TM^ 2X Fast Q-PCR Master Mix SYBR no ROX (SMOBIO Technology, Inc., Hsinchu, Taiwan) were used for RT-qPCR analysis according to the manufacturer’s instructions. The 2^−^^∆∆*CT*^ method was used to calculate the relative gene expression levels of genes of interest: three genes encoding antioxidant enzymes (*CuZnSOD1, APX2,* and *CAT3*), three genes encoding TFs (*OsMYB91, OsWRKY42*, and *OsWRKY70*), and the proline biosynthesis gene *OsP5CS1,* with *UBIQUITIN* used as an internal reference gene [60]. All primer sequences used in this study are listed in Table 3.

### 4.9. Statistical Analysis

This study used a factorial, completely randomized design with three biological replicates, except for the germination test. The data were processed using IBM SPSS statistics version 25 software. Analysis of variance (ANOVA) was carried out followed by the Duncan’s multiple range test (DMRT) between the means of the treatments to determine the significant difference at a 5% of significance level (*p* < 0.05).

## Figures and Tables

**Figure 1 plants-12-02879-f001:**
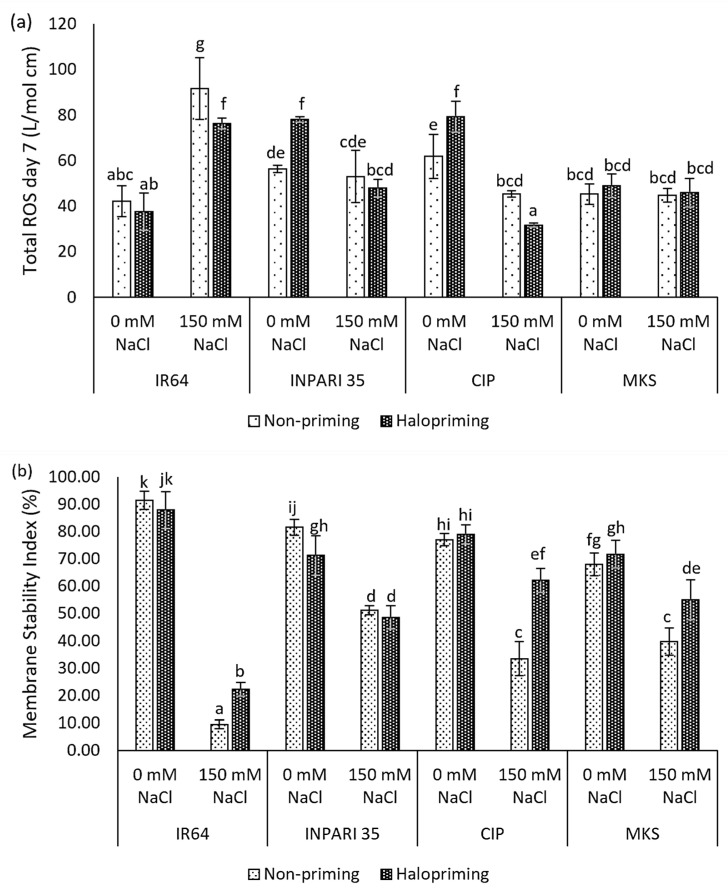
Halopriming effect on oxidative damage: total ROS levels (**a**) and membrane stability index on day 7 (**b**) of control (0 mM NaCl) and salt treatment (150 mM NaCl) in IR64, INPARI 35, Cempo Ireng Pendek (CIP), and Merah Kalimantan Selatan (MKS). Values represent means ± SD of three biological replicates. Different lowercase letters indicate significant differences (*p* < 0.05) based on DMRT.

**Figure 2 plants-12-02879-f002:**
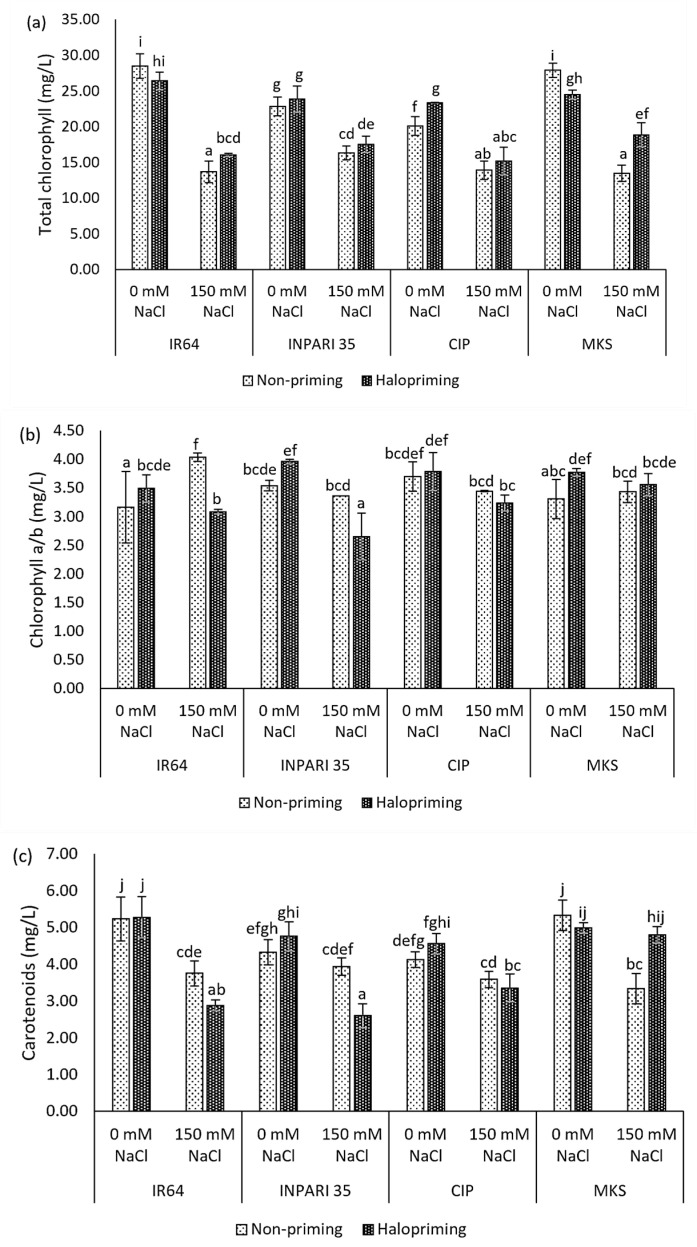
Photosynthetic pigment contents: total chlorophyll content (**a**), chlorophyll a/b ratio (**b**), carotenoids content (**c**) on day 7 of control (0 mM NaCl) and salt-stress (150 mM NaCl) treatment in IR64, INPARI 35, Cempo Ireng Pendek (CIP), and Merah Kalimantan Selatan (MKS). Values represent means ± SD of three biological replicates. Different lowercase letters indicate significant differences (*p* < 0.05) based on DMRT.

**Figure 3 plants-12-02879-f003:**
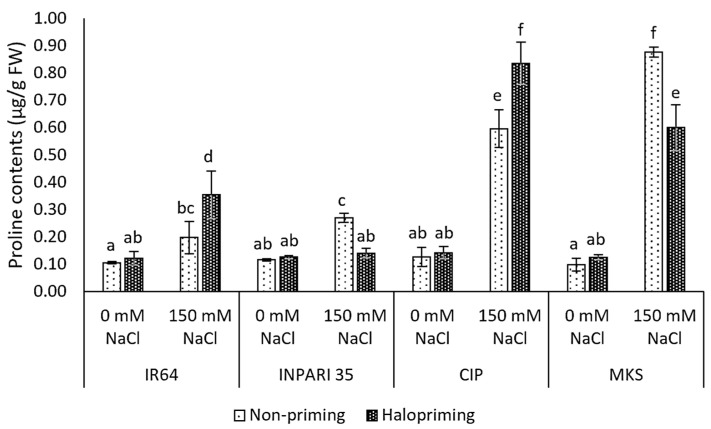
Proline contents in rice after 4 days of salt treatment. CIP, Cempo Ireng Pendek; MKS, Merah Kalimantan Selatan. Values represent means ± SD of three biological replicates. Different lowercase letters indicate significant differences (*p* < 0.05) based on DMRT.

**Figure 4 plants-12-02879-f004:**
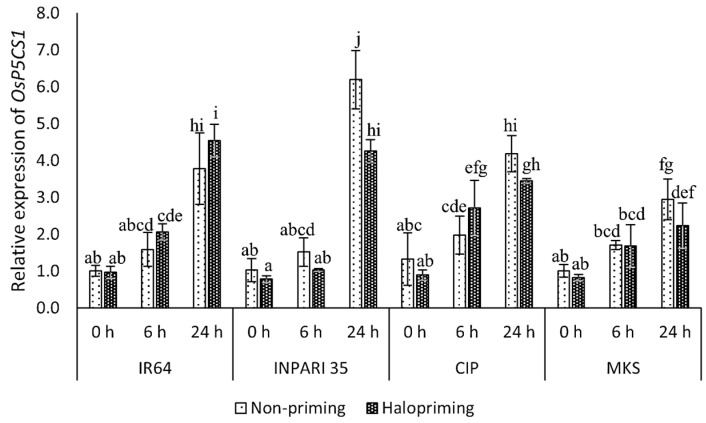
Expression analysis of *OsP5CS1* in rice under salt treatment. CIP, Cempo Ireng Pendek; MKS, Merah Kalimantan Selatan. Values represent means ± SD of three replicates. Different lowercase letters indicate significant differences (*p* < 0.05) based on DMRT.

**Figure 5 plants-12-02879-f005:**
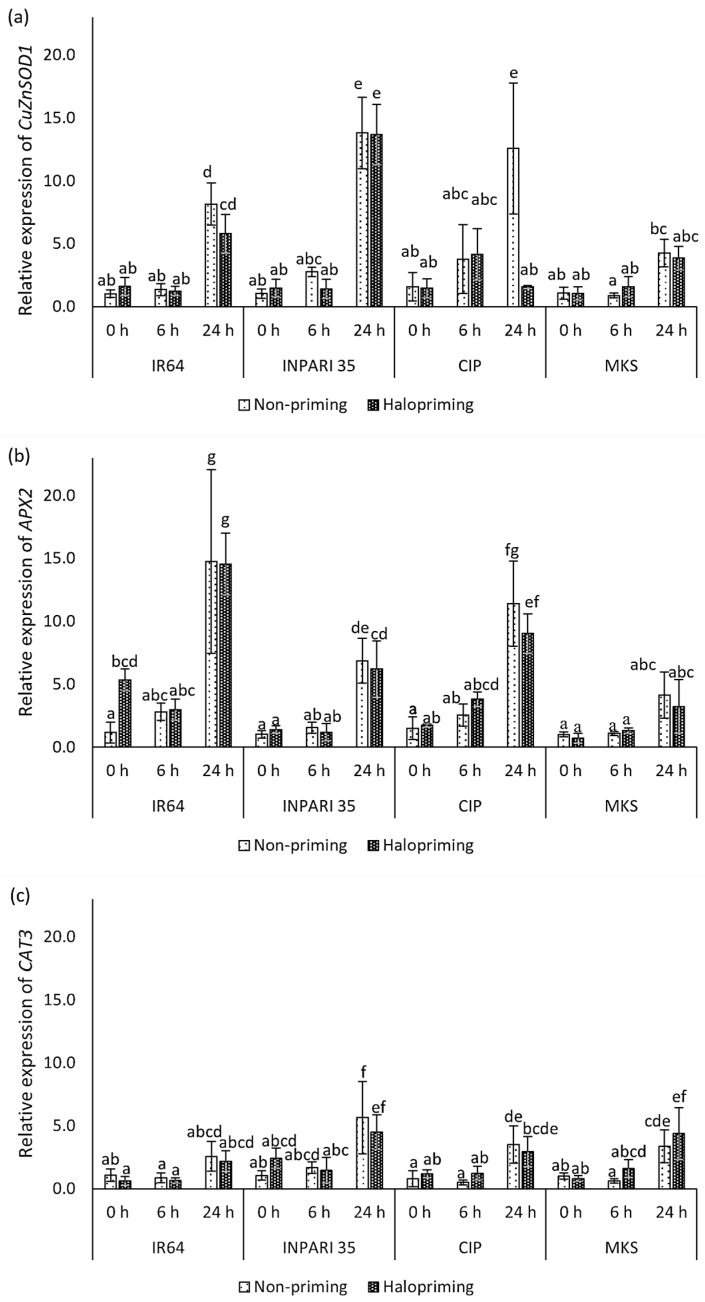
Changes in the expression of antioxidant-related genes in rice in response to halopriming and salt-stress treatment (150 Mm NaCl). Relative expression levels of *CuZnSOD1* (**a**), *APX2* (**b**)*,* and *CAT3* (**c**). CIP, Cempo Ireng Pendek; MKS, Merah Kalimantan Selatan. Values represent means ± SD of three independent replicates. Different lowercase letters indicate significant differences (*p* < 0.05) based on DMRT.

**Figure 6 plants-12-02879-f006:**
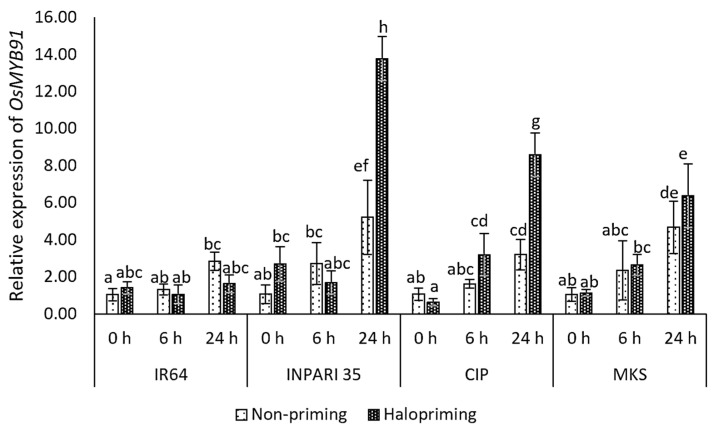
Expression of *OsMYB91* in rice under salt stress. Values represent means ± SD of three replicates. Different lowercase letters indicate significant differences (*p* < 0.05) based on DMRT.

**Figure 7 plants-12-02879-f007:**
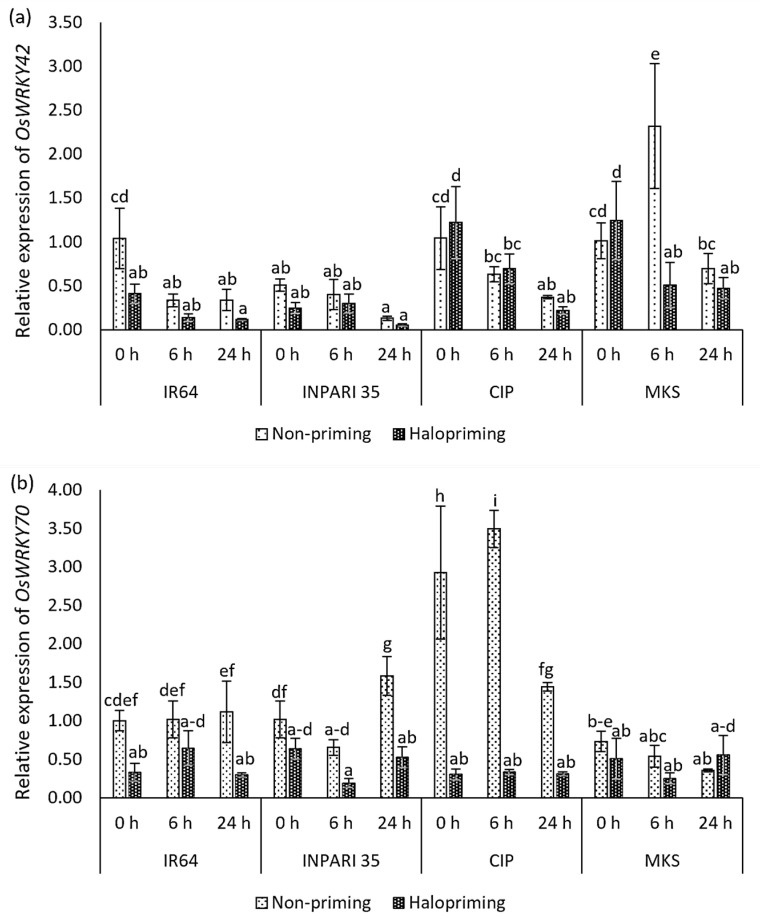
Expression of *OsWRKY42* (**a**) and *OsWRKY70* (**b**) in rice under salt stress. Values represent means ± SD of three replicates. Different lowercase letters indicate significant differences (*p* < 0.05) based on DMRT.

**Table 1 plants-12-02879-t001:** Effects of seed halopriming on germination rate and mean germination time over 7 days of soaking treatment.

Cultivar	Pre- Treatment	Germination Rate (%)/Day							MGT (Days)
		1	2	3	4	5	6	7	
IR64	NP	85 ± 10	94 ± 5	97 ± 3	100 ± 0	100 ± 0	100 ± 0	100 ± 0	1.24 ± 0.16 ^a^
	HP	63 ± 8	94 ± 7	96 ± 4	100 ± 0	100 ± 0	100 ± 0	100 ± 0	1.47 ± 0.17 ^b^
INPARI 35	NP	89 ± 4	96 ± 4	97 ± 4	100 ± 0	100 ± 0	100 ± 0	100 ± 0	1.18 ± 0.11 ^a^
	HP	91 ± 5	94 ± 2	97 ± 3	100 ± 0	100 ± 0	100 ± 0	100 ± 0	1.18 ± 0.06 ^a^
CIP	NP	12 ± 6	88 ± 3	96 ± 2	100 ± 0	100 ± 0	100 ± 0	100 ± 0	2.00 ± 0.05 ^c^
	HP	0 ± 0	87 ± 7	92 ± 3	100 ± 0	100 ± 0	100 ± 0	100 ± 0	2.21 ± 0.09 ^d^
MKS	NP	62 ± 6	98 ± 3	100 ± 0	100 ± 0	100 ± 0	100 ± 0	100 ± 0	1.40 ± 0.08 ^b^
	HP	39 ± 7	100 ± 0	100 ± 0	100 ± 0	100 ± 0	100 ± 0	100 ± 0	1.61 ± 0.07 ^c^

Mean value followed by different lowercase letters in MGT parameter indicates significant differences (*p* < 0.05) based on DMRT. CIP, Cempo Ireng Pendek; MKS, Merah Kalimantan Selatan; NP, no priming; HP, halopriming.

**Table 2 plants-12-02879-t002:** Effects of seed halopriming on seedling height and root length after 7 days of salt treatment.

Cultivar	Pre- Treatment	Plant Height (cm)		Root Length (cm)		RWC (%)	
		0 mM	150 mM	0 mM	150 mM	0 mM	150 mM
IR64	NP	34.2 ± 1.0 ^ab^	31.5 ± 0.9 ^a^	11.3 ± 0.6 ^cd^	8.5 ± 0.5 ^ab^	90.0 ± 3.51 ^cde^	81.8 ± 0.79 ^b^
	HP	35.8 ± 0.9 ^bc^	31.6 ± 1.0 ^a^	10.2 ± 0.4 ^bc^	7.8 ± 1.1 ^a^	94.8 ± 2.80 ^f^	89.9 ± 2.81 ^c^
INPARI 35	NP	47.5 ± 1.2 ^fg^	40.2 ± 1.5 ^d^	14.7 ± 1.2 ^ef^	9.3 ± 0.6 ^abc^	95.4 ± 1.09 ^f^	88.9 ± 3.32 ^cde^
	HP	50.0 ± 1.5 ^g^	40.8 ± 1.6 ^de^	15.9 ± 2.9 ^f^	9.7 ± 1.3 ^abc^	93.8 ± 1.98 ^ef^	86.5 ± 2.93 ^c^
CIP	NP	46.8 ± 0.8 ^e^	41.0 ± 1.7 ^de^	13.3 ± 1.2 ^e^	9.1 ± 0.8 ^ab^	94.5 ± 2.28 ^f^	82.4 ± 0.99 ^ab^
	HP	43.1 ± 3.5 ^de^	40.5 ± 2.2 ^de^	12.8 ± 1.5 ^de^	10.5 ± 0.2 ^bc^	95.2 ± 2.54 ^f^	80.4 ± 2.15 ^b^
MKS	NP	39.9 ± 0.5 ^d^	36.4 ± 2.3 ^bc^	8.6 ± 0.8 ^ab^	8.9 ± 0.8 ^ab^	93.3 ± 1.79 ^ef^	80.05 ± 4.29 ^ab^
	HP	36.8 ± 0.4 ^bc^	38.5 ± 0.9 ^cd^	11.3 ± 0.8 ^cd^	8.9 ± 0.4 ^ab^	92.8 ± 0.47 ^def^	76.65 ± 0.82 ^a^

Mean value followed by different lowercase letters in each parameter indicates significant differences (*p* < 0.05) based on DMRT. Data represent means ± SD of three biological replications. CIP, Cempo Ireng Pendek; MKS, Merah Kalimantan Selatan; NP, no priming; HP, halopriming.

**Table 3 plants-12-02879-t003:** Primer sequences used for gene expression analysis.

Gene	Sequence (5′ → 3′)	Amplicon Size (bp)
*CuZnSOD1*	F: GAGATTCCAAACCAGCAGGA	277
	R: TTGTAGTGTGGCCCAGTTGA	
*APX2*	F: TCTTCCTGATGCCACACAAG	298
	R: GTCCTCATCCGCAGCATATT	
*CAT3*	F: ACCGGTTCATCAAGAGATGG	304
	R: ACACGAATTGTGCGGTGATA	
*OsMYB91*	F: CCACCTCCTTTACTTGAGC	161
	R: ATCCTGCTGCTCTGTTCTT	
*OsWRKY42*	F: ACGACTGACCAAACTACTGG	158
	R: CAATTGGCAAATACTACGTG	
*OsWRKY70*	F: CGTATAGGGAGAACGAGAAA	183
	R: ATAGCAAAGCCATAGAACGA	
*OsP5CS1*	F: TTCTTGGGCATGCTGATGGT	192
	R: ATTGCAGGCTGCTGGGTAAT	
*UBQ*	F: AACCAGCTGAGGCTGATGGT	77
	R: ATTGCAGGCTGCTGGGTAAT	

## Data Availability

Not applicable.

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
