# Peer review of "Seed Halopriming: A Promising Strategy to Induce Salt Tolerance in Indonesian Pigmented Rice"

_plants, 2023, doi:10.3390/plants12152879_

Round 1
Reviewer 1 Report (New Reviewer)
The present paper “Seed halopriming: a promising strategy to induce salt tolerance 2 in Indonesian pigmented rice” by Purwestri et al. investigated the responses of pigmented rice to salt stress following seed halopriming to enhance plant salt tolerance. Seeds of 4 rice cultivars were used in this study. The work could be ameliorated and I suggest major issues:
Row 17: “crop losses worldwide” could be changed in “yield losses worldwide”. On the same line, “Salinity is a major factor affecting crop yields” could be removed.
Row 28-29: I suggest to explain what are IR64 and INPARI because at the beginning of the abstract you introduce 2 rice cultivars, not 4. For example, you can add: ….IR64 (a salt stress-sensitive cultivar) ….INPARI (salt stress-tolerant).
Row 29-30 You introduce many genes without explaining the analysis that you are doing. You can add that these are stress-related genes….
Row 32 I suggest to substitute “usefulness” with “ potential benefits”
Row 44 I suggest to substitute “meaning they” with “i.e.”
Row 57 I suggest to substitute “being” with “are”
Row 62 For instance, the expression of CuZnSOD1….
Row 70 “P5CS1 controls proline 69 biosynthesis in the chloroplast, and its expression increases under stress conditions.” Please add the citation at the end of the sentence.
Row 78 Many sentences are too short … OsMYB91 is induced by salt stress and rice plants overexpressing Os-78 MYB91 showed increased tolerance to salt stress
Row 100-107 I suggest to better specify the contribution you give with this research and the goals of this paper. Otherwise it seems that you just did it in the previous work.
Row 125-126 No references in the results. The same for Row 184-185 and Row 212. These sentences are ok for the discussion, in the results you should only comment your graphs/tables.
In the discussion you should highlight the most important results obtained in this study, without repeating ALL the results. I suggest focusing on what is important to highlight in your research.
Row 410-415: This is a repetition of what you just said. I suggest focusing on the future perspective, discussing the possibility to apply this treatment in a real system (field conditions). The applicability in the field is really important.
In material and methods (and also in the introduction) you should highlight how the genes have been selected. By what criteria were they chosen?
Can be improved
Author Response
Please see the attachment.

Reviewer 2 Report (New Reviewer)
There are some problems in the manuscript listed as follow:
1 L434, Mean times germination (MTG) was measured after 7 days of soaking. The author should provide the cited reference for the MDT method, especially for the research about seed germination, what’s meaning of the soaking, is salinity soaking or priming?
The author can read the related reference such as “Relationship between the length of the lag period of germination and the emergence performance of oat (Avena sativa) seeds, MAO P.S., ZHANG X. Y., SUN Y., ZHANG W. X., WANG Y. W., Seed Science and Technology 2013, 41,281-291” to understand the detail information about the MDT method.
2 L438, After 7 days, healthy, uniform seedlings were transferred to a greenhouse in a floating hydroponic cultivation system. According to the author described, the seedlings were selected to plant in the greenhouse, but what is meaning for the seed halopriming? Please providing related citations to support the role of halopriming on the plant growth.
3 L440, After 2 weeks, the seedlings were subjected to salt stress created by the addition of 150 mM NaCl to the cultivation solution and incubation for seven days. This is a confused point, the role of seed priming could affect the seedling growth? Furthermore, why carried out the treatment for addition of 150 mM NaCl? Please make sure the treatment of seed priming on the germination.
4 the author measured several indexes including contents of chlorophyll, carotenoids, ROS and proline, but the author selected different salt stress duration. So the design in the experiment is difficult to understand.
Author Response
Please see the attachment.

Reviewer 3 Report (New Reviewer)
The mamuscript entitled “Seed halopriming: a promising strategy to induce salt tolerance in Indonesian pigmented rice” is trying to present if seed halopriming could affect pigmented rice’s salt tolerance. The halopriming was exposed on seeds of for rice caltivars. The primed seeds were then germinated and seedlings were used in saline treatment. The data collected were mainly focused on plant growth, chlorophyll content, ROS related index and gene expressions. Such physiological investigation is interesting. However, present study has two major issues:
1. Methods part were not clearly and the data did not exhibited properly. e.g., in table 1, how many seeds and repetitions were used for germination rate and MGT? in the colum of MGT, what did the alphabet mean? what kind of statistics used and was it ok? The statistic method maybe was not right.
2. Some statement was no preciseness. For example, statement in line 128-129 “Seedling height under salt stress was slightly higher in seedlings that had undergone halopriming compared to non-primed seedling, except in Cempo Ireng Pendek” did not comply with the data showed in table 2. And there were other vague or outlying statements.
Minors
1. In abstract, line 20-21, “In this study, we investigated 20 the responses of these two pigmented rice cultivars to salt stress following seed halopriming to enhance plant salt tolerance.” “In this study, we investigated if seed halopriming enhance salt tolerance in these two pigmented rice cultivars.”
2. In abstract, line 23-24, “We pre-treated seeds with halopriming solution before germination and then exposed the plants to salt stress 150 mM NaCl at 21 days after germination….”
3. In abstract, line 25-26, “Halopriming helped mitigate the negative effects of salinity on plant growth….” “Halopriming could mitigate the negative effects of salinity on plant growth….”
4. In abstract, line 30, “Halopriming also had varying effects in the expression of stress-related genes….” “Halopriming also had disparate effects in the expression of stress-related genes….”
5. In abstract, line 30, “These findings shed light on 32 salt-tolerance mechanisms in Indonesian pigmented rice and highlight the potential benefits of 33 halopriming in salt-affected agro-ecosystems.” “These findings highlighted the potential benefits of halopriming in salt-affected agro-ecosystems.”
language in some part was not explicit and could be improved ...
Round 2
Reviewer 1 Report (New Reviewer)
now the paper is worthy of publication
Author Response
We would like to express our gratitude to Reviewer 1 for the comments and suggestions in order to improve our paper and make it worthy for publication.
Reviewer 2 Report (New Reviewer)
3 L440, After 2 weeks, the seedlings were subjected to salt stress created by the addition of 150 mM NaCl to the cultivation solution and incubation for seven days. This is a confused point, the role of seed priming could affect the seedling growth? Furthermore, why carried out the treatment for addition of 150 mM NaCl? Please make sure the treatment of seed priming on the germination.
Response:
In the first place, we did halopriming before we germinated the seeds. The observation of seed germination rate and MGT aim to make sure that halopriming solution is not harmful for the seeds. Our research investigated the effect of halopriming in response to salt stress at the seedling stage. Therefore, we treated the seedlings (21 days old) with salt stress by adding 150 mM NaCl to the hydroponic system (using Yoshida medium).
?just like the author said, there are different treatments for the seed halopriming and salt stress, so the results of seed halopriming actually wasn’t related with those of the 150 mM NaCl treatment. Furthermore, these results also didn’t support the title of seed halopriming could indue the rice seed salt tolerance.
Round 3
Reviewer 2 Report (New Reviewer)
3 L440, After 2 weeks, the seedlings were subjected to salt stress created by the addition of 150 mM NaCl to the cultivation solution and incubation for seven days. This is a confused point, the role of seed priming could affect the seedling growth? Furthermore, why carried out the treatment for addition of 150 mM NaCl? Please make sure the treatment of seed priming on the germination.
Response: Thank you for your comment. Here we try to explain.
Our research is trying to present if seed halopriming could affect pigmented rice’s salt tolerance. The halopriming was exposed on seeds of four rice caltivars and the primed seeds were then germinated, and the seedlings were used in saline treatment. Therefore, the halopriming and salt stress treatment is a consecutive process. For control, we used nonprimed seed. We revised the part of result by deleting the first sentence “To explore the effects of halopriming and salinity on seed germination and seedling growth in rice, we measured plant height and root length as indicators of plant growth” (in previous manuscript line 122-123) to avoid misunderstanding and make new subheading title:
1. Effect of seed halopriming on germination (line 108)
2. Plant growth and Relative Water Content Water (RWC) (line 123) And we added the sentence “To explore the effects of halopriming on seedling growth in rice under salt stress, we measured plant height and root length as indicators of plant growth” in the beginning of this paragraph (line 124-125)
this is my comment: Just like the author explained in the problem 3. Seed halopriming had no affection on the seed germination and seedling growth. All results only showed the effect of salt stress on the seedling. So, the role of seed halopriming actually could not be illustrated with these results, also the title cannot be supported by the contents in this manu.
The key point is seed halopriming the author did actually has no relation with salt stress, but what the author revised could not proved the presentation of the title.
Author Response
Please see the attachment.

This manuscript is a resubmission of an earlier submission. The following is a list of the peer review reports and author responses from that submission.
Round 1
Reviewer 1 Report
The manuscript has an interesting topic but all the scientific information is ruined by the misconstruction of two very important words as follow: effect and salt tolerance. As I suggested early in the title, there is a controversial affirmation, because no effect can be visible on any type of tolerance. Effect is a change being a consequence of an action while tolerance, in the case of plants, is the ability to endure continued subjection to environmental conditions without adverse reaction. Therefore, plants' responses (increased proline level, growth parameters etc.) to stress conditions define the level of their tolerance against unfavorable conditions. Halopriming as a "strategy", force plant seeds to germinate activating their survival mechanisms/defense mechanisms which subsequently lead to a clear definition of their tolerance level against salt or any other stress. The underlaying mechanisms are based exactly on those pathways enlisted in the article, involving various chemical compounds and reactions (proline, MDA, SOD, Antioxidant activity, sometimes glycine-betaine, phenols, flavonoids, total soluble sugars and so on. The increase or decrease of these compounds indicate the plant's tolerance level to a specific stress. In addition, avoid to use "salinity stress" since both words are nouns (salt stress is recommended) and correct it all through the manuscript. As a conclusion I suggest to revise these terms and make sure you use them accordingly.
Another observation is the lack of the main aim or purpose of this study. I understood it well, but needs to be described precisely and correct using the correct meaning of the terms mentioned above not only in the abstract but also at the end of the Introduction part.
Further observations are mentioned in the manuscript, please follow the comments.
Last but not least, the manuscripts needs an English check in order to avoid any misunderstanding or misuse of the scientific terms.

Reviewer 2 Report
Please see the attachment for review comments of the manuscript.
